# PIGDreamer: Privileged Information Guided World Models for Safe Partially Observable Reinforcement Learning

Dongchi Huang[1]   Jiaqi Wang[2]   Yang Li[1]   Chunhe Xia[1]   Tianle Zhang[3]   *Kaige Zhang[4]

## Abstract

Partial observability presents a significant challenge for Safe Reinforcement Learning (Safe RL), as it impedes the identification of potential risks and rewards. Leveraging specific types of privileged information during training to mitigate the effects of partial observability has yielded notable empirical successes. In this paper, we propose Asymmetric Constrained Partially Observable Markov Decision Processes (ACPOMDPs) to theoretically examine the advantages of incorporating privileged information in Safe RL. Building upon ACPOMDPs, we propose the Privileged Information Guided Dreamer (*PIGDreamer*), a model-based RL approach that leverages privileged information to enhance the agent's safety and performance through privileged representation alignment and an asymmetric actor-critic structure. Our empirical results demonstrate that *PIGDreamer* significantly outperforms existing Safe RL methods. Furthermore, compared to alternative privileged RL methods, our approach exhibits enhanced performance, robustness, and efficiency. Codes are available at: `https://github.com/hggforget/PIGDreamer`.

## 1. Introduction

Safety presents a significant challenge to the real-world applications of reinforcement learning (RL) (Feng et al., 2023; Ji et al., 2024; OpenAI et al., 2019; Papadimitriou & Tsitsiklis, 1987). Several researches (Liu et al., 2021; Thomas et al., 2022; Achiam et al., 2017) has been dedicated to addressing this challenge, relying on constrained

---
*Equal contribution  [1]School of Computer Science, University of Beihang, Beijing, China [2]School of Computer Science, Chinese University of Hong Kong, Hongkong, China [3]JD Explore Academy, Beijing, China [4]North Automatic Control Institute, Taiyuan, China. Correspondence to: Kaige Zhang <zkgusu@gmail.com>.

*Proceedings of the 42nd International Conference on Machine Learning*, Vancouver, Canada. PMLR 267, 2025. Copyright 2025 by the author(s).

Markov decision processes (CMDPs) (Altman, 2021), in which agents operate on the underlying states with the objective of maximizing rewards while ensuring that costs remain below predefined constraint thresholds. However, existing approaches overlook the inherent partial observability in real-world applications, thereby leading to the suboptimal deployment of Safe RL algorithms.

To mitigate partial observability in Safe RL, recent studies (Hogewind et al., 2022; As et al., 2022; Huang et al., 2024) have investigated the application of world models (LeCun & Courant, 2022), resulting in remarkable success. These methods are situated within a more realistic framework: Constrained Partially Observable Markov Decision Processes (CPOMDPs) (Lee et al., 2018), where the agent operates on a history of previous observations and actions, without direct access to the underlying states. Thus, world models (Hafner et al., 2023) which learn environmental dynamics and task-specific predictions from past observations and actions, have been shown to effectively memorize historical information and enhance sample efficiency. However, despite enabling agents to utilize historical information, these models introduce computational (Papadimitriou & Tsitsiklis, 1987) and statistical (Krishnamurthy et al., 2016; Jin et al., 2020) chanllengs under partial observations.

To address these challenges, recent studies (Hu et al., 2024; Lambrechts et al., 2024) investigate methods for leveraging privileged information within world models. The utilization of privileged information is a practical solution, as in the real-world deployment of RL, only a subset of sensors is accessible during testing (Hu et al., 2024) while a greater number of sensors is available during training. Furthermore, for agents trained in simulators and subsequently transferred to the real world through Sim2Real approaches (Yamada et al., 2023; Pinto et al., 2017), the underlying states in the simulators can be effectively leveraged to enhance the agents. Despite their remarkable empirical successes, these approaches still encounter two challenges: (1) The theoretical understanding of leveraging privileged information in the world models remain relatively limited (Cai et al., 2025); (2) These approaches exhibit limited efficiency in leveraging privileged information (Hu et al., 2024). These issues raise a crucial question:

*How can we achieve safe, high-performance, theoretically guaranteed and easily trainable agents under conditions of partial observability by leveraging privileged information?*

To address these problems, we propose a novel framework Asymmetric Constrained Partially Observable Markov Decision Processes (ACPOMDPs), elucidating the mechanisms for integrating privileged information. Our theoretical analysis within this framework demonstrates that agents will underestimate potential risks in the absence of privileged information. Furthermore, building upon our framework, we present Privileged Information Guided Dreamer (`PIGDreamer`), which enhances the world models, predictors, representations, and critics with privileged information generally (i.e., the underlying states, past actions, and proprioceptive sensors). We integrate `PIGDreamer` with the Lagrangian method (Nocedal & Wright, 2006; Li et al., 2021) to ensure the satisfaction of safety constraints. Experiments on the Safety-Gymnasium (Ji et al., 2023b) and Guard (Zhao et al., 2024) benchmarks demonstrated that `PIGDreamer` surpasses existing Safe RL methods in both safety and performance. Furthermore, `PIGDreamer` achieves a 136% improvement in performance with only a 28% increase in training time, illustrating superior utilization of privileged information compared to alternative privileged RL methods.

Our key contributions are summarized as follows:

- We introduce a novel theoretical framework, ACPOMDPs, wherein we theoretically demonstrate that asymmetric inputs decrease the number of critic updates and facilitate the development of a more optimal policy.

- Built upon ACPOMDPs, we present `PIGDreamer`, a safe, high-performance, and efficient algorithm that leverages privileged information in model-based RL.

- Our empirical results across diverse benchmarks indicate that `PIGDreamer` surpasses both existing safe RL methods and privileged model-based RL methods.

## 2. Preliminaries

**CPOMDPs**  Sequential decision making problems under partial observations are typically formulated as Partially Observable Markov Decision Processes (POMDPs) (Egorov et al., 2017), represented as the tuple $(S, A, P, R, Z, O, \gamma)$. The state space is denoted as $S$ and the action space as $A$. The transition probability function is denoted as $P(s'|s, a)$. Let $Z$ be the observation space, $O(z|s', a)$ stands for the observation probability. The reward function is represented by $R(s, a)$. The discount factor is represented by $\gamma$. In POMDPs framework, the agent has access only to the observations $z_t$ and actions $a_t$ at each time step $t$, without direct knowledge of the underlying state of the environment.

As a result, the agent must maintain a belief state $b_t$, where $b_t(s) = Pr(s_t = s|h_t, b_0)$ represents the probability distribution over possible states $s$, given the history $h_t = \{z_0, a_0, z_1, a_1, \ldots, a_{t-1}, z_t\}$ of past actions and observations, and the initial belief state $b_0$. We denote the set consisting of all possible belief states as $B$, the belief reward function as $R_B(b, a) = \sum_{s \in S} b(s) R(s, a)$, the transition function as $\tau(b, a, z)$. For simplicity, we write $\tau(b, a, z)$ as $b^{a,z}$. Finally, let the policy be denoted as $\pi_\theta(a \mid b)$. The objective of the policy is to maximize the long-term belief expected reward $V_R(b_0)$ as follows:

$$\max_\pi V_R(b_0) = E_{a_t \sim \pi}[\textstyle\sum_{t=0}^{\infty} \gamma^t R_B(b_t, a_t) \mid b_0], \quad (1)$$

Constrained POMDPs (CPOMDPs) is a generalization of POMDPs defined by $(S, A, P, R, Z, O, C, d, \gamma)$. The cost function set $C = \{(C_i, b_i)\}_{i=1}^m$ comprises individual cost functions $C_i$ and their corresponding cost thresholds $b_i$. The goal is to compute an optimal policy that maximizes the long-term belief expected reward $V_R(b_0)$ while bounding the long-term belief expected costs $V_{C_i}(b_0)$ as follows:

$$\max_\pi V_R(b_0) = E_{a_t \sim \pi}[\textstyle\sum_{t=0}^{\infty} \gamma^t R_B(b_t, a_t) \mid b_0],$$
$$s.t. V_{C_i}(b_0) = E_{a_t \sim \pi}[\textstyle\sum_{t=0}^{\infty} \gamma^t C_{iB}(b_t, a_t) \mid b_0] \le b_i, \forall i \in [m] \quad (2)$$

where in practical implementations, $V_R(b_0)$ is updated by the following Bellman optimal equation:

$$V_R^*(b) = \max_{a \in A} \left[ R_B(b, a) + \gamma \sum_{z \in Z} Pr(z|b, a) V_R^*(b^{a,z}) \right], \quad (3)$$

and the $V_{C_i}(b_0)$ is updated equivalently.

**Safe RL with Model-based Methods**  Model-based RL approaches (Moerland et al., 2022; Polydoros & Nalpantidis, 2017; Berkenkamp et al., 2017) provide significant advantages for addressing Safe RL problems by facilitating the modeling of environmental dynamics. Jayant & Bhatnagar (2022); Thomas et al. (2022) enhance the integration of model-free Safe RL algorithms with dynamic models by employing ensemble Gaussian models. Alternatively, Zanger et al. (2021) utilize neural networks (NNs) to quantify model uncertainty and constrain this error measure through constrained model-based policy optimization. Recently, As et al. (2022) has integrated Bayesian methods with the Dreamer framework (Hafner et al., 2020) to quantify uncertainty in the estimated model, employing the Lagrangian method to incorporate safety constraints. However, these works overlook the issue of partial observability, which presents a more realistic setting in real-world applications. In this context, Hogewind et al. (2022) addresses the problem of partial observability in Safe RL by integrating the Lagrangian mechanism into the SLAC framework (Lee et al., 2020). Huang et al. (2024) achieved zero-cost performance by integrating the Lagrangian method with the

Dreamerv3 framework. From the perspective of POMDPs (Kaelbling et al., 1998), constructing world models solely from partial observations does not fully exploit the potential of these models. In this work, we aim to address partial observability by leveraging privileged information that is accessible during training, thereby ensuring the agent's safety and enhancing its performance.

**Privileged RL with World Models**  While extensively studied in the literature on model-free RL (Pinto et al., 2017; Salter et al., 2021; Baisero & Amato, 2021; Baisero et al., 2022; Li et al., 2020), leveraging privileged information is seldom addressed in model-based RL. Yamada et al. (2023) represent the first attempt to utilize privileged information in the training of world models, employing it by distilling the learned latent dynamics model from the teacher to the student world model. Since the underlying state transitions of privileged information and partial observations differ, direct model distillation may eliminate essential features from partial observations. Lambrechts et al. (2024) incorporate privileged information solely by introducing an auxiliary objective that predicts the privileged information exclusively. Inspired by scaffolding teaching mechanisms in psychology, Hu et al. (2024) developed an approach to exploit privileged sensing in critics, world models, reward estimators, and other auxiliary components used exclusively during training, achieving outstanding empirical results. However, this approach adds too many components, resulting in low training efficiency. Avalos et al. (2024) presents a method for approximating belief updates, supported by theoretical assurances that the resulting beliefs enhance the learning process of the optimal value function. In summary, the current exploration of world models that utilize privileged information remains insufficient. Consequently, we aim to identify the optimal model architecture that efficiently and robustly incorporates privileged information within world models.

# 3. Asymmetric Constrained Partially Observable Markov Decision Processes (ACPOMDPs)

In this section, we introduce our formulation of the Asymmetric Constrained Partially Observable Markov Decision Processes (ACPOMDPs) and compare it with the standard CPOMDPs. This comparison highlights the advantages of utilizing an asymmetric architecture.

## 3.1. Framework Setup

We propose ACPOMDPs, a relaxed variant of CPOMDPs. The key distinction is that ACPOMDPs assumes the availability of the underlying states when computing the long-term expected values. Thus, ACPOMDPs are formulated by a tuple $(S, B, A, \tau, R_B, C, d, \gamma)$, and aims to maximize the

long-term belief expected reward $V_R(b)$ while bounding the long-term belief expected costs $V_{C_i}(b)$:

$$
\begin{aligned}
\pi_\star &= \arg\max_{a \in A} V_R(b). \\
&s.t. V_{C_i}(b) \leq b_i, \forall i \in [m]
\end{aligned}
\tag{4}
$$

Benefiting from the availability of the underlying states, at each time step, the critic receives and the action $a$ and the underlying state $s$, and updates the $V_R^*(s)$ using the following equation:

$$
V_R^*(s) = \max_{a \in A} \left[ R(s, a) + \gamma \sum_{s'} P(s'|s, a) V_R^*(s') \right].
\tag{5}
$$

Consequently, the $V_R(b)$ and $V_{C_i}(b)$ in the optimization problem (4) are estimated using this updated $V_R^*(s)$:

$$
V_R^*(b) = \sum_{s \in S} b(s) V_R^*(s).
\tag{6}
$$

Notice that the update of the $V_{C_i}(b)$ is not presented, which is equivalent to (6).

## 3.2. Comparison with CPOMDPs

We compare the different estimations of the $V_R(b)$ and $V_C(b)$ to demonstrate the superiority of ACPOMDPs. Since the $V_R(b)$ and $V_C(b)$ are equivalent in their estimations, we Collectively refer to them as $V(b)$. For clarity, we rewrite the $V(b)$ in CPOMDPs as $V_{sym}(b)$ and $V(b)$ in ACPOMDPs as $V_{asym}(b)$.

**Lemma 3.1.** *Kaelbling et al. (1996) showed that the value function at time step $t$ can be expressed by a set of vectors: $\Gamma_t = \{\alpha_0, \alpha_1, \ldots, \alpha_m\}$. Each $\alpha$-vector represents an $|S|$-dimensional hyper-plane, and defines the value function over a bounded region of the belief:*

$$
V_t^*(b) = \max_{\alpha \in \Gamma_t} \sum_{s \in S} \alpha(s) b(s).
\tag{7}
$$

**Lemma 3.2.** *Assume the state space $S$, action space $A$, and observation space $Z$ are finite. Let $|S|$, $|A|$, and $|Z|$ represent the number of states, actions, and observations, respectively. Let $|\Gamma_{t-1}|$ denote the size of the solution set for the value function $V_{t-1}(b)$ at time step $t-1$. The minimal number of elements required to express the value function $V_t(b)$ at time step $t$, denoted as $|\Gamma_t|$, grows as $|\Gamma_t| = O(|A||\Gamma_{t-1}|^{|Z|})$ (Pineau et al., 2006).*

We conclude that, at each time step $t$, the belief state space can be represented as a discrete representation space that exactly captures the value function $V_t(b)$. The size of this space is given by $|\Gamma_t| = O(|A||\Gamma_{t-1}||Z|)$. Furthermore, as derived from (6), in the ACPOMDPs framework, the

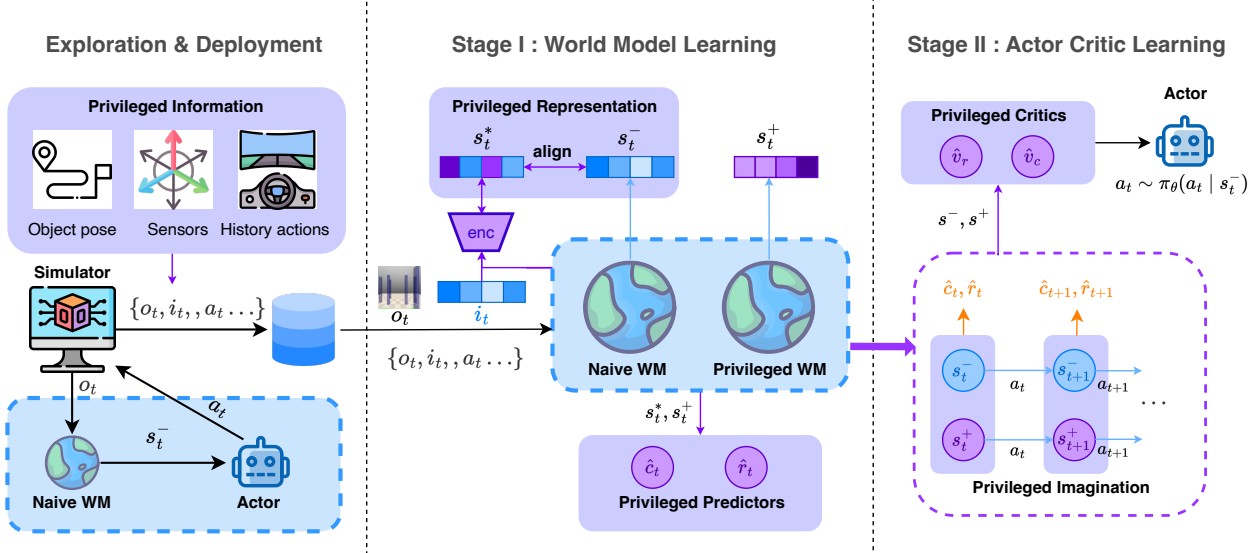

*Figure 1.* Overview of PIGDreamer. The components highlighted in purple are trained using privileged information, which is collected from the simulators during exploration. We enhance the estimation of the critics by granting access to privileged information, thereby improving the actor's policy. Additionally, privileged representations are constructed to provide the actor with more information.

required size of the representation space is reduced to $|S|$. Clearly, $|S| \ll |\Gamma_t| = O(|A||\Gamma_{t-1}|^{|Z|})$.

Thus, ACPOMDPs significantly reduce the size of the representation space required to express the value function $V(b)$, eliminating observation-related uncertainties to the greatest extent possible. This, in turn, reduces the number of updates required for the critic to estimate the value function $V(b)$.

**Theorem 3.3.** *Let $V_{asym}^*(b)$ and $V_{sym}^*(b)$ represent the optimal long-term expected values under the ACPOMDPs and CPOMDPs frameworks, respectively. Then, for all belief states $b \in B$, the inequality holds: $V_{asym}^*(b) \geq V_{sym}^*(b)$. (The proof is provided in Appendix A)*

The conclusion indicates that ACPOMDPs, by leveraging asymmetric information, yields superior policies compared to CPOMDPs. This is because, for any belief state $b \in B$, the optimal long-term expected reward under ACPOMDPs is always greater than or equal to that under CPOMDPs. Regarding safety, ACPOMDPs provide more accurate estimations due to additional information, while long-term expected costs under CPOMDPs are consistently lower or equal to those of ACPOMDPs. This implies that CPOMDPs tend to underestimate future safety risks.

## 4. Methodology

Grounded in the ACPOMDP framework, we now present PIGDreamer, a Safe RL approach that utilizes privileged information to improve the policy of an agent operating under partial observability.

### 4.1. Overview

We base our model on DreamerV3 (Hafner et al., 2023), a renowned algorithm noted for its cross-domain generality and insensitivity to hyperparameters, wherein the world models are implemented as a Recurrent State-Space Model (RSSM) (Hafner et al., 2019). In our framework, we incorporate additional components trained using privileged information to enhance policy training, subsequently disabling these components during deployment to ensure that the policy operates solely on partial observations. Specifically, during training, we concurrently develop two world models that serve as environment simulators: the naive world model and the privileged world model. The naive world model learns from the agent's observations, while the privileged world model learns from the underlying state. Subsequently, the actor and the critics are trained using the abstract sequences generated by these world models, where the actor operates exclusively on partial observations, while the critics are granted access to privileged information. Finally, during deployment, only the naive world model and the actor, which operates on partial observations, are activated. Figure 1 provides a clear visualization of our training pipeline. In summary, we leverage privileged information in three distinct ways:

**Privileged Representations.** Intuitively, providing the actor with a more informative representation results in a superior policy. Inspired by this concept, we enhance the representations within the naive world model by aligning them with those derived from privileged information.

**Privileged Predictors.** Accurate prediction of rewards and

costs from partial observations is infeasible, as they do not contain comprehensive information about the underlying states. We address this challenge by providing the predictors with privileged information.

**Privileged Critics.** Based on our Theorem 3.3, granting critics access to privileged information can enhance policy performance. Consequently, we provide the critics with privileged information to refine their estimations.

### 4.2. Privileged Information Guided World Model

The world models are parameterized by learnable network parameters $\phi$. Each world model operates on its model state $s$. At each time step $t$, the world models receive an observation $o_t$, an action $a_t$, and privileged information $i_t$ as inputs. Encoders map $o_t$ and $i_t$ to embeddings $z_t^-$ and $z_t^+$, respectively. The dynamics predict the next states $\hat{s}_{t+1}$ based on the action $a_t$. The posteriors utilize embeddings $z_t^-$ and $z_t^+$ and the predicted states $\hat{s}_{t+1}$, to predict the true underlying states $s_{t+1}$. Finally, reward and cost predictors use the concatenation of $s_t^*$ and $s_t^+$ to predict rewards and costs, while decoders employ their corresponding $s_t^*$ and $s_t^+$ to reconstruct the inputs $o_t$ and $i_t$. Detailed definitions of these components are provided below:

Naive World Model

$$
\begin{cases}
\text{Encoder: } p_\phi(z_t^- \mid o_t) \\
\text{Decoder: } p_\phi(\hat{o}_t, \hat{i}_t \mid s_t^*) \\
\text{Dynamics: } p_\phi(\hat{s}_t^- \mid s_{t-1}^-, a_{t-1}) \\
\text{Posterior: } q_\phi(s_t^- \mid \hat{s}_t^-, z_t^-) \\
\text{Oracle Posterior: } q_\phi(s_t^* \mid \hat{s}_t^-, z_t^-, z_t^+)
\end{cases}
$$

Privileged World Model

$$
\begin{cases}
\text{Encoder: } p_\phi(z_t^+ \mid i_t) \\
\text{Decoder: } p_\phi(\hat{i}_t \mid s_t^+) \\
\text{Dynamics: } p_\phi(\hat{s}_t^+ \mid s_{t-1}^+, a_{t-1}) \\
\text{Posterior: } q_\phi(s_t^+ \mid \hat{s}_t^+, z_t^+) \\
\text{Reward Predictor: } p_\phi(\hat{r}_t \mid s_t^*, s_t^-) \\
\text{Cost Predictor: } p_\phi(\hat{c}_t \mid s_t^*, s_t^-)
\end{cases}
$$

$$(8)$$

The world models are classified into two categories: the naive world model and the privileged world model. The naive world model, which relies solely on the observation $o_t$ as input, is available to agents. As depicted in the Figure 2, the world models predicts state transitions using the dynamics by minimizing the dynamic loss between the predicted states $\hat{s}_t$ and the true underlying states $s_t$. The dynamic loss is presented in Equation 9.

$$\mathcal{L}_{dyn} = \mathcal{L}_{rep}(\hat{s}_t^-, s_t^-) + \mathcal{L}_{rep}(\hat{s}_t^+, s_t^+), \quad (9)$$

$$\mathcal{L}_{rep}(q, p) = \alpha \, \mathrm{KL}\left[q \parallel \mathrm{sg}(p)\right] + \beta \, \mathrm{KL}\left[\mathrm{sg}(q) \parallel p\right], \quad (10)$$

in the above, $\mathrm{sg}(\cdot)$ represents the stop-gradient operator, while $\mathrm{KL}[\cdot]$ denotes the Kullback-Leibler (KL) divergence.

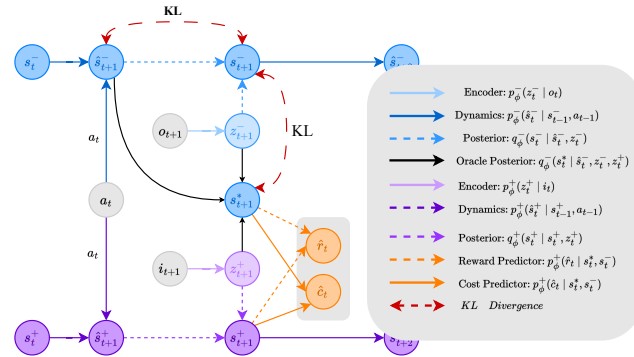

*Figure 2.* Visualization of the world model learning pipeline. Blue represents the components belonging to the naive world model, while purple indicates the components associated with the privileged world model.

**Representation Alignment.** Specially, in the naive world model, at each time step $t$, the oracle posterior predicts the oracle representations $s_t^*$, which encapsulate both the observation $o_t$ and the privileged information $i_t$. We enforce the alignment between the states $s^-$ and the oracle states $s^*$ with the align loss defined in Equation 11 to enhance the state representations of the naive world model as follows:

$$\mathcal{L}_{align} = \mathcal{L}_{rep}(s_t^*, s_t^-). \quad (11)$$

**Privileged Predictors.** In the PIG World Model, the predictors predict rewards and costs utilizing the most informative state representations $s^*$ and $s^+$. Here, $s^*$ encompasses the observational information upon which the agent operates, while $s^+$ contains the privileged information that enhances prediction capabilities. The predictors are optimized with the following loss:

$$\mathcal{L}_{pred} = \underbrace{-\ln p_\phi^+(\hat{r}_t \mid s_t^*, s_t^-)}_{\text{reward loss}} \underbrace{-\ln p_\phi^+(\hat{r}_t \mid s_t^*, s_t^-)}_{\text{cost loss}}. \quad (12)$$

Afterwards, the decoders are supervised to reconstruct their corresponding inputs using the reconstruction loss in Equation (13). Specifically, the decoder in the naive world model reconstructs the inputs $o_t$ and $i_t$ using the oracle representations $s_t^*$ to capture all relevant information. This information is then distilled into $s_t^-$. Compared to previous works (Lambrechts et al., 2024; Hu et al., 2024) that directly reconstruct privileged information $i_t$ from the state representations $s_t^-$, this method is significantly more robust when the privileged information is excessively informative for reconstruction:

$$\mathcal{L}_{dec} = \underbrace{-\ln p_\phi^-(\hat{o}_t, \hat{i}_t \mid s_t^*)}_{\text{observation}} \underbrace{-\ln p_\phi^+(\hat{i}_t \mid s_t^+)}_{\text{privileged information}}. \quad (13)$$

Finally, the objective of the privileged information guided world model can be summarized as follows:

$$\mathcal{L}_\phi = \mathcal{L}_{dyn} + \mathcal{L}_{align} + \mathcal{L}_{dec} + \mathcal{L}_{pred}. \quad (14)$$

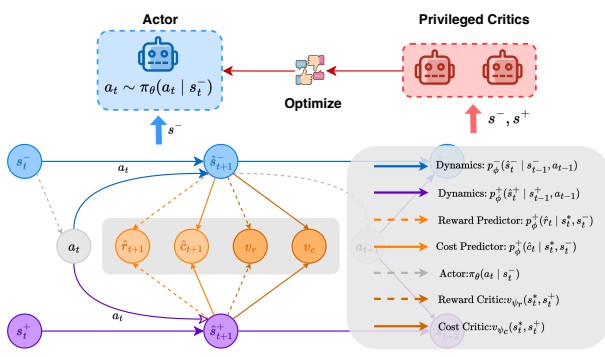

*Figure 3.* Visualization of the actor critic learning pipeline. The lower part presents the generation process of the abstract sequences, dubbed *Twisted Imagination*. The upper part demonstrates the learning mechanism of the actor-critic model.

### 4.3. Privileged Safe Actor Critic Learning

The actor and critics learn purely from the abstract sequences generated by world models. Specifically, at time step $t$, the actor, parameterized with the learnable network parameter $\theta$, operates on the state $s_t^-$ to predict the policy distribution $\pi_\theta(a_t \mid s_t^-)$. The critics, on the other hand, operate on the state $s_t^-$ and $s_t^+$ to estimate the long-term expected returns $v_{\psi_r}(s_t^-, s_t^+)$ and $v_{\psi_r}(s_t^-, s_t^+)$. Figure 3 is presented for a clear visualization of our training pipeline. In summary, the key components of the actor and critics are:

$$
\begin{aligned}
\text{Actor:} &\quad a_t \sim \pi_\theta(a_t \mid s_t^-) \\
\text{Reward Critic:} &\quad v_{\psi_r}(s_t^-, s_t^+) \approx \mathbb{E}_{\pi_\theta}\left[R_t^\lambda\right] \quad (15)\\
\text{Cost Critic:} &\quad v_{\psi_c}(s_t^-, s_t^+) \approx \mathbb{E}_{\pi_\theta}\left[C_t^\lambda\right]
\end{aligned}
$$

**Twisted Imagination (TI).** *Twisted Imagination* is a procedure for generating abstract sequences from the world model, leveraging low-dimensional privileged information. As illustrated in Figure 3, starting from the representations of replayed inputs $s_t^-$ and $s_t^+$, the actor samples an action $a_t$ from the policy distribution $\pi_\theta(a_t \mid s_t^-)$, utilizing $s_t^-$ from the naive world model. Subsequently, each world model predicts its next representations $s_{t+1}^-$ and $s_{t+1}^+$. The cost $\hat{c}_t$ and reward $\hat{r}_t$ can then be predicted by the respective predictors until the time step $t$ reaches the imagination horizon $H = 15$. This synchronization of the imagination process across the two world models enables the actor and critics to learn from coherent simulated trajectories.

**Privileged Critics.** After the *Twisted Imagination*, an abstract trajectory $\left\{s_t^-, s_t^+, a_t, \hat{r}_t, \hat{c}_t\right\}_{1:H}$ is provided to the actor and the critics. Similar to privileged predictors, the critics are supplied with privileged information $s_t^+$ and the state $s_t^-$ to generate more accurate estimations. Specifically, based on the abstract trajectory, the critics can estimate the long-term expected returns $v_{\psi_r}(s_t^-, s_t^+)$ and $v_{\psi_c}(s_t^-, s_t^+)$,

while the actor optimizes its policy according to a specified objective. From the given imaginary trajectory, the bootstrapped $\text{TD}(\lambda)$ value $R^\lambda(s_t)$ for the reward critic is calculated as follows:

$$
R^\lambda(s_t^-, s_t^+) = \hat{r}_t + \gamma((1-\lambda)V_{\psi_r}\left(s_{t+1}^-, s_{t+1}^+\right) \quad (16)
$$
$$
+ \lambda R^\lambda(s_t^-, s_t^+)),
$$
$$
R^\lambda(s_T^-, s_T^+) = V_{\psi_r}\left(s_T^-, s_T^+\right). \quad (17)
$$

These values are used to assess the long term expected reward, where $V_{\psi_r}(s_t)$ is approximated by the reward critic to consider the returns that beyond the imagination horizon $H$. Note that, we show here only the calculation of $R^\lambda(s_t)$, the calculation of $C^\lambda(s_t)$ is equivalent to (16).

**Lagrangian Constrained Policy Optimization.** With the calculated $\text{TD}(\lambda)$ values $R^\lambda(s_t)$ and $C^\lambda(s_t)$, we follow Equation (27) to define the policy optimization objective:

$$
\mathcal{L}(\theta) = -\sum_{t=1}^{T} \text{sg}\left(R^\lambda(s_t)\right) + \eta \text{H}\left[\pi_\theta\left(a_t \mid s_t^-\right)\right]
$$
$$
- \underbrace{\Psi\left(C^\lambda(s_t), \lambda_k^p, \mu_k\right)}_{\text{penalty term}}. \quad (18)
$$

This policy optimization objective encourages the actor to maximize the expected reward while simultaneously satisfying the specified safety constraints. The penalty term is formulated using the Augmented Lagrangian method (Dai & Zhang, 2021), which penalizes behaviors that violate safety constraints. Additionally, an entropy term is included in the objective to promote exploration. Further details regarding the policy optimization objective and the Augmented Lagrangian method can be found in Appendix C.

## 5. Experiments

**Experimental Setup.** In all our experiments, the agent observes a 64x64 pixel RGB image from the onboard camera. We assess the task objective performance and safety using the following metrics:

- Average undiscounted episodic return for $E$ episode: $\hat{J}(\pi) = \frac{1}{E}\sum_{i=1}^{E}\sum_{t=0}^{T_{\text{ep}}} r_t$.

- Average undiscounted episodic cost return for $E$ episode: $\hat{J}_c(\pi) = \frac{1}{E}\sum_{i=1}^{E}\sum_{t=0}^{T_{\text{ep}}} c_t$.

We compute $\hat{J}(\pi)$ and $\hat{J}_c(\pi)$ by averaging the sum of costs and rewards across $E = 10$ evaluation episodes of length $T_{ep} = 1000$. The results for all methods are recorded once the agent reached $4M$ environment steps. Detailed designs of privileged information, descriptions of all baselines, and additional experiments can be found in Appendix D.

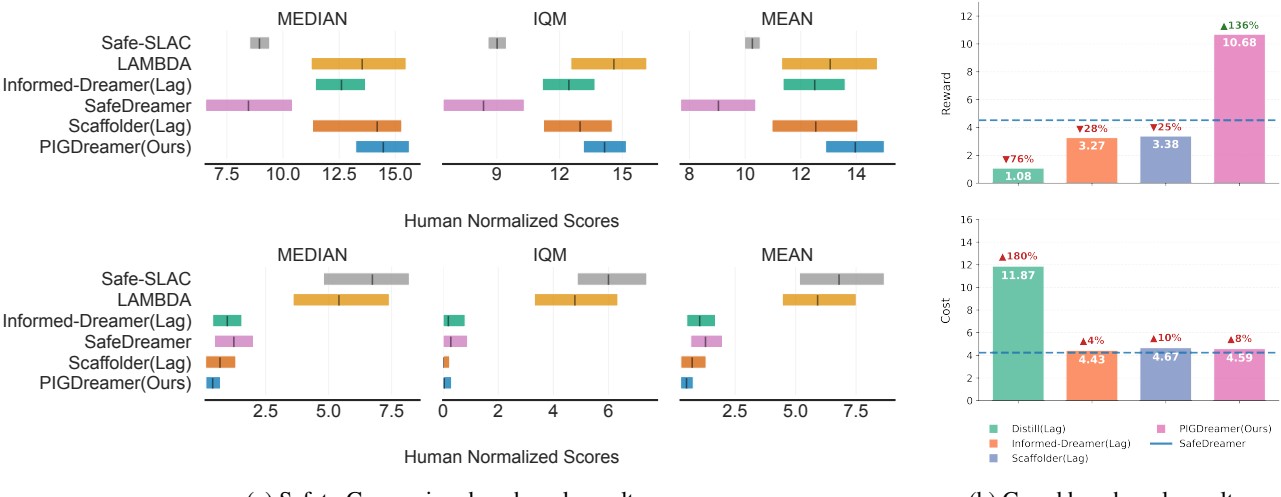

(a) Safety Gymnasium benchmark results.

(b) Guard benchmark results.

*Figure 4.* Aggregate metrics for summarizing benchmark performance. In Figure 4a, we employ the rliable library (Agarwal et al., 2021) to compute the median, inter-quartile mean (IQM), and mean estimates for normalized reward and cost returns, accompanied by 95% bootstrap confidence intervals (CIs) based on three runs across five tasks. In Figure 4b, we calculate the average rewards and costs across three tasks to compare the performance of various privileged variants of SafeDreamer.

**Results.** As illustrated in Figure 4, our algorithm achieves state-of-the-art performance by leveraging privileged information. In terms of safety, *PIGDreamer* not only meets the safety constraints at convergence, but also attains near-zero-cost performance. Concerning rewards, *PIGDreamer* outperforms all methods, especially its unprivileged variant, SafeDreamer, which demonstrates significant improvements. Conversely, as depicted in Figure 6, LAMBDA matches *PIGDreamer* in rewards but fails to ensure safety and does not produce a practical policy for the *PointButton1* task. Additionally, Safe-SLAC performs worse, failing in the *Point-Push1* and *RacecarGoal1* tasks and exhibiting considerable safety constraint violations. As illustrated in Figure 4a, Scaffolder (Lag) and Informed-Dreamer (Lag) exhibit superior performance compared to SafeDreamer. This finding indicates that the utilization of privileged information enhances the agent's capacity to learn a more effective policy, even with limited observations, which aligns with Theorem 3.3. However, as depicted in Figure 4b, these algorithms experience a decline in performance relative to SafeDreamer after incorporating privileged information. This paradox arises because, the agent is unable to compensate for the information gap pertaining to the privileged information, resulting in sub-optimal policies. In contrast to alternative methods, *PIGDreamer* consistently demonstrates superior performance. This underscores the robustness of our approach in utilizing privileged information.

**Benchmarking Privileged Variants.** We now examine the privileged variants: Scaffolder (Lag), Informed-Dreamer (Lag), Distill (Lag), and *PIGDreamer*. The comparisons presented in Figure 4 reveal that different approaches to

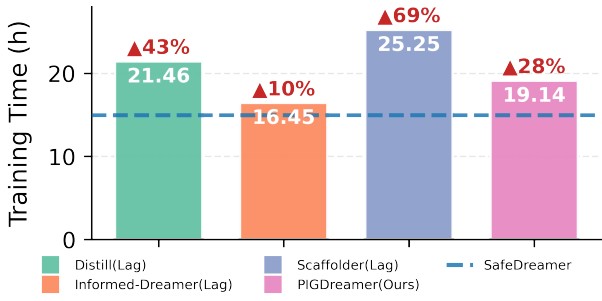

*Figure 5.* Training Efficiency Comparison: We visualize the average training time for the tasks on the Guard benchmark.

exploiting privileged information yield disparate results. Distill (Lag), which trains a privileged teacher policy and distills the knowledge to the student policy, suffers from severe performance degradations resulting from the information gap between the privileged information and partial observations. Informed-Dreamer, which relies solely on reconstructing privileged information from partial observations, achieves only a marginal performance improvement. In contrast, Scaffolder builds on Informed-Dreamer by providing predictors and critics with privileged access and incorporating an additional privileged actor for exploration, leading to significantly improved outcomes. *PIGDreamer* consistently achieves state-of-the-art performance across all benchmarks. In particular, in the Guard benchmark, *PIG-Dreamer* outperforms alternative variants with a 136% performance improvement. Nonetheless, these enhancements result in extended training times. Our analysis in Figure.5 demonstrates that Scaffolder requires an average training

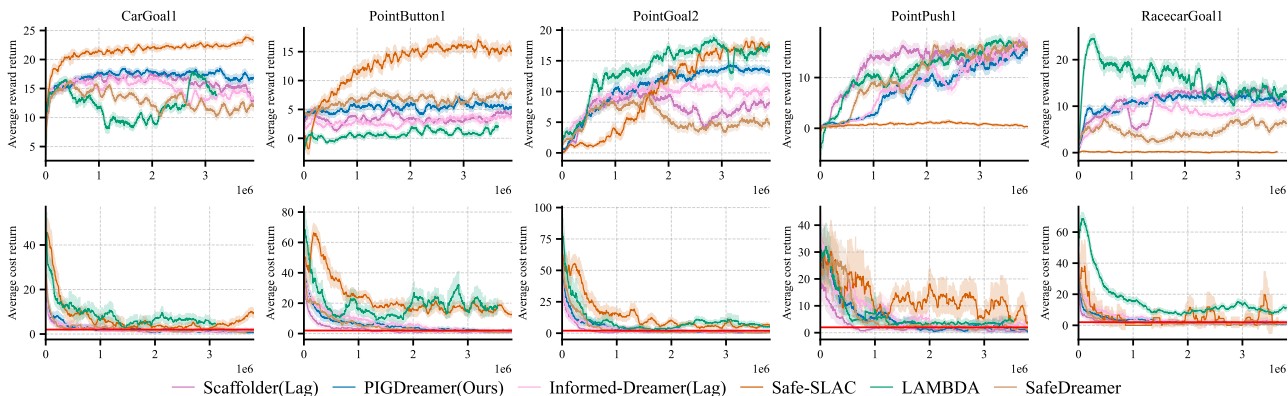

*Figure 6.* The experimental results for the Safety-Gymnasium Benchmark. The upper figures illustrate the curves of episodic reward, while the lower figures depict the curves of episodic cost. The red solid line indicates the cost constraint for this task.

time that is 69% longer compared to SafaDreamer, while Informed-Dreamer necessitates only 10% longer average training time. Our algorithm, PIGDreamer, achieves the best outcomes with only a 28% increase in average training time. This indicates that our algorithm exhibits the highest efficiency in utilizing privileged information.

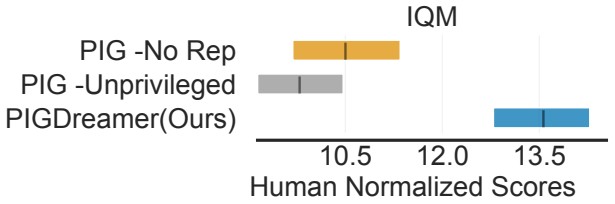

*Figure 7.* The ablation study results.

**Ablation Study.** Now we assess the component-wise contributions of the privileged components in our algorithm. Our ablation study includes the following settings: (1) **PIG -No Rep:** In this variant, the naive world model does not align the state representations with the oracle representation. (2) **PIG -Unprivileged:** In this variant, no privileged information is accessible during training. (3) **NLI:** In this variant, we replace *Twisted Imagination* with Nested Latent Imagination (NLI). Figure.7 presents the results of the ablation study conducted on the Safety-Gymnasium Benchmark, while detailed experimental results can be referenced in Figure.9. As illustrated in Figure.7, **PIG - No Rep**, shows only marginal improvement compared to **PIG - Unprivileged**. This limited enhancement occurs because the privileged critic mainly improves the agent's policy by accurately estimating the value function, which guides decision-making but is insufficient for substantial improvements. Conversely, PIGDreamer achieves significant improvements by utilizing privileged representations. This is because the privileged representations enable the agent to predict privileged information from its observations, allowing the agent to operate

on more informative representations.

| Task | Method | Reward | Cost |
|------|--------|--------|------|
| SafetyPointGoal2 | NLI | 10.79 | **0.41** |
|  | **TI (Ours)** | **13.59** | 0.73 |
| SafetyCarGoal1 | NLI | 14.79 | 0.64 |
|  | **TI (Ours)** | **17.32** | **0.43** |
| SafetyRacecarGoal1 | NLI | **13.99** | 1.57 |
|  | **TI (Ours)** | 11.38 | **0.83** |

*Table 1.* Comparison with Nested Latent Imagination (NLI).

Table 1 presents a comparison of various trajectory generation procedures. The results indicate that TI achieves competitive performance compared to NLI, while utilizing a lighter model design, which is significantly important for robustness and versatility.

## 6. Conclusion

We propose **PIGDreamer**, a model-based RL approach specifically proposed for partially observable environments with safety constraints. This approach is formalized within the *ACPOMDPs* framework, providing theoretical advantages in addressing the challenges of partial observability and safety. By leveraging privileged information through privileged representation alignment and an asymmetric actor-critic structure, *PIGDreamer* achieves competitive performance using vision-only input on the diverse tasks. Our experiments show that *PIGDreamer* is the most effective approach for utilizing privileged information within world models, excelling in performance, safety and efficiency. We believe *PIGDreamer* marks a significant advancement in harnessing RL for real-world applications. However, we noted that privileged information did not always lead to improvements in certain tasks, necessitating further research to explore the relationships between specific types of privileged information and particular tasks.

## Acknowledgement

This work was supported by the National Natural Science Foundation of China (Grant No. 62272024).

## Impact Statement

This paper presents work whose goal is to advance the field of Machine Learning. There are many potential societal consequences of our work, none which we feel must be specifically highlighted here.

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

## A. Proof

In this section, we prove Theorem 3.3. First, we donate the optimal long-term values in the belief space under the ACPOMDPs framework as $V_{asym}^*(b)$:

$$V_{asym}^*(b) = \sum_{s \in S} b(s) V^*(s)$$

$$V^*(s) = \max_{a \in A} \left[ R(s,a) + \gamma \sum_{s'} P(s'|s,a) V^*(s') \right]$$

(19)

where $V^*(s)$ represents the optimal long-term values in the state space. Similar to $V_{asym}^*(b)$, the optimal long-term values in the belief space under the CPOMDPs framework are represented as $V_{sym}^*(b)$:

$$V_{sym}^*(b) = \max_{a \in A} \left[ R(b,a) + \gamma \sum_{z \in Z} Pr(z|b,a) V_{sym}^*(b^{a,z}) \right]$$

(20)

Since

$$Pr(z|b,a) = \sum_{s' \in S} Pr(z|a,s') \sum_{s \in S} Pr(s'|s,a) b(s)$$

(21)

$V_{sym}^*(b)$ can be rewritted as:

$$V_{sym}^*(b) = \max_{a \in A} \left[ R(b,a) + \gamma \sum_{z \in Z} V_{sym}^*(b^{a,z}) \sum_{s' \in S} Pr(z|a,s') \sum_{s \in S} P(s'|s,a) b(s) \right]$$

$$= \max_{a \in A} \left[ R(b,a) + \gamma \sum_{z \in Z} V_{sym}^*(b^{a,z}) \sum_{s \in S} \sum_{s' \in S} Pr(z|a,s') P(s'|s,a) b(s) \right]$$

(22)

*Proof.* According to Lemma 3.1, $V^*(s')$ can be expressed by a set of vectors: $\Gamma_t = \{\alpha_0, \alpha_1, \ldots, \alpha_m\}$. As a result, $V^*(s')$ can be rewrite as the following equation:

$$V^*(s) = \max_{a \in A} \left[ R(s,a) + \gamma \sum_{s'} P(s'|s,a) \max_{\alpha \in \Gamma_t} \alpha(s') \right]$$

(23)

Similarly, $V_{sym}^*(b)$ can be rewritten as:

$$V_{sym}^*(b) = \max_{a \in A} \left[ \sum_{s \in S} R(s,a) b(s) + \gamma \sum_{z \in Z} \max_{\alpha \in \Gamma_{t-1}} \sum_{s \in S} \sum_{s' \in S} Pr(z|a,s') P(s'|s,a) b(s) \alpha(s') \right]$$

(24)

Then:

$$V_{asym}^*(b) = \sum_{s \in S} b(s) V^*(s)$$

$$= \sum_{s \in S} b(s) \max_{a \in A} \left[ R(s,a) + \gamma \sum_{s'} P(s'|s,a) \max_{\alpha \in \Gamma_t} \alpha(s') \right]$$

$$\geq \max_{a \in A} \left[ \sum_{s \in S} b(s) R(s,a) + \gamma \sum_{s \in S} b(s) \sum_{s' \in S} P(s'|s,a) \max_{\alpha \in \Gamma_t} \alpha(s') \right]$$

$$= \max_{a \in A} \left[ R(b,a) + \gamma \sum_{s \in S} \sum_{s' \in S} \sum_{z \in Z} Pr(z|a,s') P(s'|s,a) b(s) \max_{\alpha \in \Gamma_t} \alpha(s') \right]$$

$$\geq \max_{a \in A} \left[ R(b,a) + \gamma \sum_{z \in Z} \max_{\alpha \in \Gamma_t} \sum_{s \in S} \sum_{s' \in S} Pr(z|a,s') P(s'|s,a) b(s) \alpha(s') \right]$$

$$= V_{sym}^*(b)$$

(25)

# B. Hyperparameters

## B.1. PIGDreamer & SafeDreamer & Informed-Dreamer & Scaffolder

*Table 2.* Hyperparameters

| Name | Symbol | Value |
|---|---|---|
| **World Model** | | |
| Number of latent classes | | 48 |
| Classes per latent | | 48 |
| Batch size | $B$ | 64 |
| Batch length | $T$ | 16 |
| Learning rate | | $10^{-4}$ |
| Coefficient of KL divergence in loss | $\alpha_q, \alpha_p$ | 0.1, 0.5 |
| Coefficient of decoder in loss | $\beta_o, \beta_r, \beta_c$ | 1.0, 1.0, 1.0 |
| **Planner** | | |
| Planning horizon | $H$ | 15 |
| Number of samples | $N_{\pi N}$ | 500 |
| Mixture coefficient | $M$ | 0.05 |
| $N_{\pi\theta} = M \cdot N_{\pi N}$ | | |
| Number of iterations | $J$ | 6 |
| Initial variance | $\sigma_0$ | 1.0 |
| **PID Lagrangian** | | |
| Proportional coefficient | $K_p$ | 0.01 |
| Integral coefficient | $K_i$ | 0.1 |
| Differential coefficient | $K_d$ | 0.01 |
| Initial Lagrangian multiplier | $\lambda_{p0}$ | 0.0 |
| Lagrangian upper bound | | 0.75 |
| Maximum of $\lambda_p$ | | |
| **Augmented Lagrangian** | | |
| Penalty term | $\nu$ | $5^{-9}$ |
| Initial penalty multiplier | $\mu_0$ | $1^{-6}$ |
| Initial Lagrangian multiplier | $\lambda_{p0}$ | 0.01 |
| **Actor Critic** | | |
| Sequence generation horizon | | 15 |
| Discount horizon | $\gamma$ | 0.997 |
| Reward lambda | $\lambda_r$ | 0.95 |
| Cost lambda | $\lambda_c$ | 0.95 |
| Learning rate | | $3 \cdot 10^{-5}$ |
| **General** | | |
| Number of other MLP layers | | 5 |
| Number of other MLP layer units | | 512 |
| Train ratio | | 512 |
| Action repeat | | 4 |

## B.2. Safe-SLAC

Hyperparameters for Safe-SLAC. We maintain the original hyperparameters unchanged, with the exception of the action repeat, which we adjust from its initial value of 2 to 4.

*Table 3.* Hyperparameters for Safe-SLAC

| Name | Value |
| --- | --- |
| Length of sequences sampled from replay buffer | 15 |
| Discount factor | 0.99 |
| Cost discount factor | 0.995 |
| Replay buffer size | $2 \times 10^5$ |
| Latent model update batch size | 32 |
| Actor-critic update batch size | 64 |
| Latent model learning rate | $1 \times 10^{-4}$ |
| Actor-critic learning rate | $2 \times 10^{-4}$ |
| Safety Lagrange multiplier learning rate | $2 \times 10^{-4}$ |
| Action repeat | 4 |
| Cost limit | 2.0 |
| Initial value for $\alpha$ | $4 \times 10^{-3}$ |
| Initial value for $\lambda$ | $2 \times 10^{-2}$ |
| Warmup environment steps | $60 \times 10^3$ |
| Warmup latent model training steps | $30 \times 10^3$ |
| Gradient clipping max norm | 40 |
| Target network update exponential factor | $5 \times 10^{-3}$ |

## B.3. LAMBDA

Hyperparameters for LAMBDA. We maintain the original hyperparameters unchanged, with the exception of the action repeat, which we adjust from its initial value of 2 to 4.

*Table 4.* Hyperparameters for LAMBDA

| Name | Value |
| --- | --- |
| Sequence generation horizon | 15 |
| Sequence length | 50 |
| Learning rate | $1 \times 10^{-4}$ |
| Burn-in steps | 500 |
| Period steps | 200 |
| Models | 20 |
| Decay | 0.8 |
| Cyclic LR factor | 5.0 |
| Posterior samples | 5 |
| Safety critic learning rate | $2 \times 10^{-4}$ |
| Initial penalty | $5 \times 10^{-9}$ |
| Initial Lagrangian | $1 \times 10^{-6}$ |
| Penalty power factor | $1 \times 10^{-5}$ |
| Safety discount factor | 0.995 |
| Update steps | 100 |
| Critic learning rate | $8 \times 10^{-5}$ |
| Policy learning rate | $8 \times 10^{-5}$ |
| Action repeat | 4 |
| Discount factor | 0.99 |
| TD($\lambda$) factor | 0.95 |
| Cost limit | 2.0 |
| Batch size | 32 |

## C. The Augmented Lagrangian

The Augmented Lagrangian method incorporates the safety constraints into the optimization process by adding a penalty term to the objective function. This allows the actor model to optimize the expected reward while simultaneously satisfying the specified safety constraints. As a result, by adopting the Augmented Lagrangian method, we transform the optimization problem in (4) into an unconstrained optimization problem:

$$\max_{\pi \in \Pi} \min_{\boldsymbol{\lambda} \geq 0} \left[ R(\pi) - \sum_{i=1}^{C} \lambda^i \left( C_i(\pi) - b_i \right) + \frac{1}{\mu_k} \sum_{i=1}^{C} \left( \lambda^i - \lambda_k^i \right)^2 \right] \tag{26}$$

where $\lambda^i$ are the Lagrange multipliers, each corresponding to a safety constraint measured by $C_i(\pi)$, and $\mu_k$ is a non-decreasing penalty term corresponding to gradient step $k$. We take gradient steps of the following unconstrained objective:

$$\tilde{R}(\pi; \boldsymbol{\lambda}_k, \mu_k) = R(\pi) - \sum_{i=1}^{C} \Psi(C_i(\pi), \lambda_k^i, \mu_k) \tag{27}$$

We define $\Delta_i = C_i(\pi) - b_i$. The update rules for the penalty term $\Psi(C_i(\pi), \lambda_k^i, \mu_k)$ and the Lagrange multipliers $\lambda^i$ are as follow:

$$\forall i \in [m] : \Psi(C_i(\pi), \lambda_k^i, \mu_k), \lambda_{k+1}^i = \begin{cases} \lambda_k^i \Delta_i + \frac{\mu_k}{2} \Delta_i^2, \lambda_k^i + \mu_k \Delta_i & \text{if } \lambda_k^i + \mu_k \Delta_i \geq 0 \\ -\frac{\left(\lambda_k^i\right)^2}{2\mu_k}, 0 & \text{otherwise.} \end{cases} \tag{28}$$

## D. Experiments

### D.1. Privileged Information Design

In different tasks, it is necessary to customize the use of different privileged information, and different privileged information will have different impacts, we show our privileged information settings in our experiments.

| Privileged Information Name | Dimension | Description |
|---|---|---|
| hazards | $(n, 3)$ | Represents the relative positions of hazards in the environment, containing 3D coordinates $[x, y, z]$. |
| velocimeter | $(3, )$ | Provides the agent's velocity information in three-dimensional space $[v_x, v_y, v_z]$. |
| accelerometer | $(3, )$ | Provides the agent's acceleration information in three-dimensional space $[a_x, a_y, a_z]$. |
| gyro | $(3, )$ | Provides the agent's angular velocity information $[\omega_x, \omega_y, \omega_z]$. |
| goal | $(3, )$ | Represents the relative coordinates of the target position that the agent needs to reach $[x_{\text{goal}}, y_{\text{goal}}, z_{\text{goal}}]$. |
| robot_m | $(3, 3)$ | Represents the rotation matrix of the robot, describing the robot's orientation and rotation in three-dimensional space. |
| past_1_action | $(4, )$ | Represents the action information from the previous time step. |
| past_2_action | $(4, )$ | Represents the action information from the second-to-last time step. |
| past_3_action | $(4, )$ | Represents the action information from the third-to-last time step. |
| euler | $(2, )$ | Represents the agent's pose information given in Euler angles $[roll, pitch]$. |

*Table 5.* Privileged Information: SafetyQuadrotorGoal1

### D.2. Experimental Setup

**Setup.** Our experiments were conducted using the following configuration: a single A100-PCIE-40GB GPU (40GB), a 10 vCPU Intel Xeon Processor (Skylake, IBRS), and 72GB of memory.

**Baselines.** We compared PIGDreamer to several competitive baselines to demonstrate the superior results of using privileged information. The baselines include: 1. **Scaffolder(Lag):** (Hu et al., 2024) Integrates Scaffolder with the Lagrangian methods. 2. **Informed-Dreamer(Lag):** (Lambrechts et al., 2024) Integrates Informed-Dreamer with the Lagrangian methods. 3. **SafeDreamer:** (Huang et al., 2024) Integrates Dreamerv3 with the Lagrangian methods. 4. **LAMBDA:** (As et al., 2022) A novel model-based approach utilizes Bayesian world models and the Lagrangian methods. 5. **Safe-SLAC:** (Hogewind et al., 2022) Integrates SLAC with the Lagrangian methods. 6. **CPO:** Achiam et al. (2017) the first general-purpose policy search algorithm for constrained reinforcement learning with guarantees for near-constraint satisfaction at each iteration. 7. **PPO_Lag:** Achiam & Amodei (2019) Integrates PPO with the Lagrangian methods. 8. **TRPO_Lag:** Integrates TRPO with the Lagrangian methods. 9. **FOCOPS:** Zhang et al. (2020) initially determines the optimal update policy by addressing a constrained optimization problem within the nonparameterized policy space. Subsequently, FOCOPS projects the update policy back into the parametric policy space. Notably, the OSRP, OSRP_Lag and SafeDreamer are three algorithms proposed by SafeDreamer.

### D.3. Addition Experiments

| | GoalAnt8Hazards | | | GoalAnt8Ghosts | | | GoalHumanoid8Hazards | | | Average | | |
|---|---|---|---|---|---|---|---|---|---|---|---|---|
| Methods | Reward | Cost | Time | Reward | Cost | Time | Reward | Cost | Time | Reward | Cost | Time |
| SafeDreamer | 2.54 | 1.00 | 14.27 | 10.67 | 0.93 | 16.7 | 0.34 | 10.8 | 13.94 | 4.52 | 4.24 | 14.97 |
| Distill-Dreamer(Lag) | 1.32 | 10.52 | 20.12 | 0.50 | 1.02 | 23.21 | 1.41 | 24.07 | 21.04 | 1.08 | 11.87 | 21.46 |
| Informed-Dreamer(Lag) | 1.32 | 0.03 | 17.03 | 10.09 | 2.54 | 18.28 | -1.60 | 10.72 | 14.03 | 3.27 | 4.43 | 16.45 |
| Scaffolder(Lag) | 1.86 | 0.04 | 24.82 | 6.15 | 0.83 | 27.14 | 2.13 | 13.14 | 23.8 | 3.38 | 4.67 | 25.25 |
| **PIGDreamer(Ours)** | 14.18 | 0.92 | 18.45 | 15.76 | 2.28 | 20.86 | 2.09 | 10.56 | 18.12 | 10.68 | 4.59 | 19.14 |

*Table 6.* The experimental results for the Guard Benchmark.

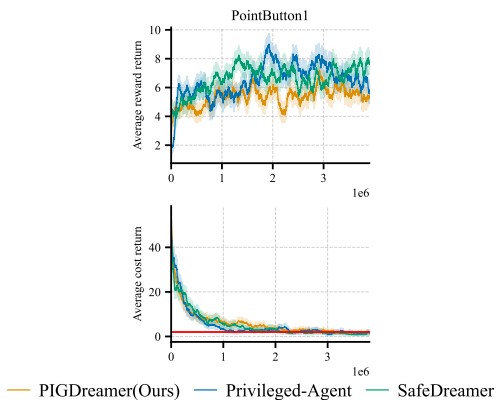

*Figure 8.* Privileged Agent in PointButton1 Task.

**Discussion** Although Figure 8 demonstrates the superior performance of exploiting privileged information, it does not achieve significant improvements in certain tasks. To provide more detail regarding our experiments, Figure 6 presents the learning curves of all algorithms. Notably, we observe that PIGDreamer does not outperform SafeDreamer in the *PointButton1* task. We conducted further experiments to investigate this phenomenon, and our findings indicate that even when the agent is granted access to privileged information, it still fails to achieve higher performance. Consequently, this finding reveals that, in the *PointButton1* task, which involves moving objects in the environment, such privileged information does not yield any information gain.

### D.4. model-free

We also compare our results with several model-free algorithms. In our experiments, the baseline algorithm results are derived from SafePO Ji et al. (2023a), which is configured with a cost limit of 25, whereas PIGDreamer is set with a cost limit of 2. The comparison results are presented in the Table 7.

| | CPO | | FOCOPS | | PPO_Lag | | TRPO_Lag | | **PIGDreamer(Ours)** | |
|---|---|---|---|---|---|---|---|---|---|---|
| Tasks | Reward | Cost | Reward | Cost | Reward | Cost | Reward | Cost | Reward | Cost |
| CarGoal1 | **23.2±1.9** | 28.2±4.6 | 21.5±0.0 | 28.1±0.0 | 13.8±3.3 | 23.4±10.8 | 22.2±3.9 | 26.2±6.1 | 14.5±0.5 | **4.2±1.2** |
| PointButton1 | 6.8±1.6 | 29.8±6.1 | 8.9±10.7 | **10.2±4.5** | 4.0±1.4 | 28.2±13.8 | 7.5±1.4 | 26.3±6.0 | **9.6±3.5** | 12.5±2.3 |
| PointPush1 | 4.8±0.0 | 25.5±0.0 | 0.7±0.7 | 23.0±21.1 | 0.6±0.3 | 26.2±25.1 | 0.6±0.1 | 21.7±11.2 | **15.6±0.8** | **0.4±0.2** |
| RacecarGoal1 | 10.4±1.2 | 29.4±7.0 | 4.5±2.2 | 93.7±33.3 | 2.3±2.1 | 28.3±12.7 | 9.5±3.0 | 25.1±5.7 | **18.2±1.2** | **6.8±1.2** |
| Average | 11.3 | 28.3 | 8.9 | 38.8 | 5.2 | 26.6 | 9.9 | 24.9 | **14.5** | **6.0** |

*Table 7.* Comparison with model-free algorithms

As illustrated in Table 7, the model-free algorithm encounters difficulties in achieving higher rewards, even with a more relaxed cost threshold. This challenge arises partly from the inability of these algorithms to leverage historical information,

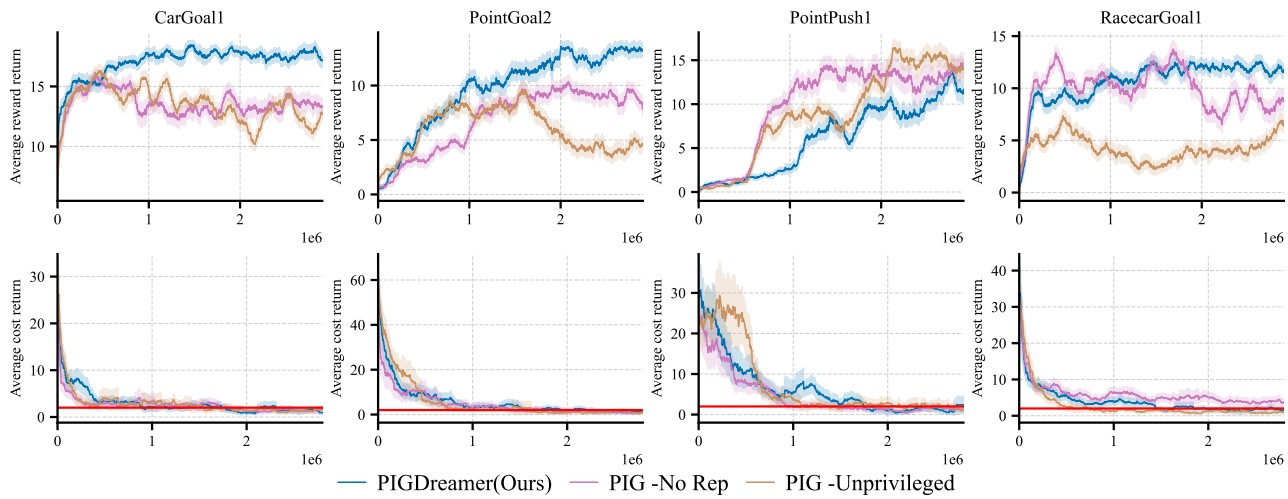

*Figure 9.* Aggregate metrics for summarizing benchmark performance. We utilize the rliable library (Agarwal et al., 2021) to compute the median, inter-quartile mean (IQM), and mean estimates for normalized reward and cost returns, accompanied by 95% bootstrap confidence intervals (CIs) based on three runs across five tasks.

as well as their lack of access to essential privileged information. In contrast, PIGDreamer significantly outperforms the baseline algorithm in both reward and safety, owing to its effective utilization of privileged information.

### D.5. Training efficiency

Since our model requires modeling two world models, we will compare the training efficiencies of the different algorithms to determine if the addition of a world model results in a significant increase in training time. Table 8 displays the hours

| | LAMBDA | Safe-SLAC | Informed-Dreamer(Lag) | Scaffolder(Lag) | SafeDreamer | **Ours** | **Incremental** |
|---|---|---|---|---|---|---|---|
| CarGoal1 | 34.1 | 18.3 | 25.77 | 45.12 | 25.51 | 36.02 | 41.2% |
| PointButton1 | 28.4 | 31.4 | 31.61 | 51.02 | 29.42 | 34.59 | 17.6% |
| PointGoal2 | 31.1 | 22.5 | 31.51 | 50.68 | 29.52 | 34.52 | 16.9% |
| PointPush1 | 36.7 | 18.3 | 25.80 | 43.87 | 23.51 | 29.43 | 25.2% |
| RacecarGoal1 | 34.1 | 20.7 | 31.08 | 49.54 | 29.59 | 33.84 | 14.3% |
| Average | 33.2 | 17.7 | 29.16 | 48.05 | 27.51 | 33.68 | 22.4% |

*Table 8.* Comparison of Training Time

required to train all algorithms for 2 million iterations on each task. The comparison between SafeDreamer and PIGDreamer reveals that the inclusion of an additional world model in PIGDreamer resulted in an average increase of 22.4% in training hours.

| | SafetyPointGoal2 | | SafetyCarGoal1 | | SafetyRacecarGoal1 | | SafetyPointPush1 | | SafetyPointButton1 | |
|---|---|---|---|---|---|---|---|---|---|---|
| Methods | Reward | Cost | Reward | Cost | Reward | Cost | Reward | Cost | Reward | Cost |
| Distill | 6.59 | 4.54 | 11.74 | 3.08 | 10.21 | 3.47 | 12.26 | 12.37 | 6.77 | 11.12 |
| Ours | 11.61 | 1.31 | 17.37 | 0.93 | 10.99 | 1.17 | 17.10 | 1.15 | 5.97 | 2.01 |

*Table 9.* Comparison of Safety Methods

