# OpenReview forum: "PIGDreamer: Privileged Information Guided World Models for Safe Partially Observable Reinforcement Learning"
_ICML.cc/2025/Conference — ICML 2025 poster_

### Official Review · Reviewer_Rrit · 2025-02-21

**Overall Recommendation:** 3

**Summary:**

This paper introduces PIGDreamer, a novel model-based reinforcement learning approach designed to enhance safety and performance in partially observable environments by leveraging privileged information during training. The authors propose Asymmetric Constrained Partially Observable Markov Decision Processes (ACPOMDPs), a theoretical framework that extends CPOMDPs by allowing the value function to access underlying states, thereby reducing the representation space and improving policy learning. PIGDreamer integrates privileged information through three key mechanisms: privileged representations, which align the state representations of the naive world model with privileged information; privileged predictors, which enhance the accuracy of reward and cost predictions; and privileged critics, which refine policy estimations. Empirical evaluations on the Safety-Gymnasium benchmark demonstrate that PIGDreamer significantly outperforms existing methods in terms of safety and task performance, achieving near-zero-cost performance while maintaining high rewards. The approach also exhibits superior training efficiency, with only a modest increase in training time compared to its unprivileged variant. Overall, PIGDreamer represents a significant advancement in model-based safe reinforcement learning for partially observable environments, effectively utilizing privileged information to achieve high performance and safety.

**Claims And Evidence:**

The claims made in the submission are generally well-supported by clear and convincing evidence.

**Essential References Not Discussed:**

N/A

**Experimental Designs Or Analyses:**

I did not find any obvious issues in the experimental designs or analyses in the paper.

**Methods And Evaluation Criteria:**

Yes, the paper's proposed methods and evaluation criteria are well-suited for the problem and application.

**Other Comments Or Suggestions:**

N/A

**Other Strengths And Weaknesses:**

**Strengths**

The paper introduces the Asymmetric Constrained Partially Observable Markov Decision Processes (ACPOMDPs) framework, which is a novel extension of existing POMDPs. This framework allows the value function to access underlying states, leading to more efficient critic updates and superior policies. This theoretical contribution is significant and provides a new perspective on leveraging privileged information in partially observable environments.

The paper demonstrates strong empirical results on the Safety-Gymnasium benchmark, showing that PIGDreamer outperforms existing methods in terms of both safety and task performance. Additionally, the method achieves these improvements with only a modest increase in training time, making it a practical solution for real-world applications.

**Weaknesses**

The paper could benefit from a more comprehensive discussion of related work, particularly recent advancements in safe RL that also leverage privileged information. This would provide a broader context for the contributions and highlight the unique aspects of PIGDreamer.

The paper evaluates PIGDreamer on the Safety-Gymnasium benchmark, but additional experiments on more diverse benchmarks or complex environments would strengthen the claims. This would provide further evidence of the method's robustness and generalizability.

While the paper reports aggregate metrics and confidence intervals, a more detailed statistical analysis, including significance tests, would further strengthen the empirical results. This would provide additional confidence in the robustness of the findings.
These points highlight the key strengths and areas for improvement in the paper, providing a balanced view of its contributions and potential enhancements.

**Questions For Authors:**

Could you elaborate on the assumptions made in the ACPOMDPs framework, particularly regarding the nature of the privileged information and its impact on the generalizability of the results? How do these assumptions affect the applicability of your framework to real-world scenarios where privileged information might not always be available or reliable?

In addition to the Safety-Gymnasium benchmark, have you considered evaluating PIGDreamer on other benchmarks or real-world datasets? If so, could you share any preliminary results or insights from those evaluations?

**Relation To Broader Scientific Literature:**

World Models: The use of world models to enhance sample efficiency and enable agents to learn from partial observations has been explored in several works (Hafner et al., 2019; Hogewind et al., 2022). These models learn environmental dynamics and task-specific predictions from past observations and actions.

The idea of using privileged information during training to improve performance has been explored in model-free RL (Hu et al., 2024; Lambrechts et al., 2024). However, its application in model-based RL, especially for safety-critical tasks, is less explored.

**Theoretical Claims:**

I did not find any obvious issues in the proofs for theoretical claims in the paper.

---

> ### Author Rebuttal · Authors · 2025-04-01
>
> We thank Reviewer Rrit for your thoughtful comments and questions. Your insights significantly contribute to the further enhancement of our paper.
>
> **Q1. A more comprehensive discussion of related work.**
>
> Thank you for your suggestion. In the next version of our manuscript, we will discuss recent advancements in safe reinforcement learning that utilize privileged information.
>
> **Q2. Guard Benchmark.**
>
> We appreciate your valuable suggestion. We conducted additional experiments within the Guard benchmark. Our empirical results indicate that our method significantly outperforms other baseline approaches. A more comprehensive response is available in **Q3** of our reply to Reviewer ou54.
>
> | Model |  | Goal_Ant_8Hazards |  |  | Goal_Ant_8Ghosts |  |  | Goal_Humanoid_8Hazards |  |
> | --- | --- | --- | --- | --- | --- | --- | --- | --- | --- |
> |  | Training time | Cost | Reward | Training time | Cost | Reward | Training time | Cost | Reward |
> | SafeDreamer | 14.27 | 1.00 | 2.54 | 16.7 | 0.93 | 10.67 | 13.94 | 10.8 | 0.34 |
> | Informed-Dreamer(Lag) | 17.03 | **0.03** | 1.32 | 18.28 | 2.54 | 10.09 | 14.03 | 10.72 | -1.60 |
> | Scaffolder(Lag) | 24.82 | 0.04 | 1.86 | 27.14 | **0.83** | 6.15 | 23.8 | 13.14 | **2.13** |
> | Distill | 20.12 | 10.52 | 1.32 | 23.21 | 1.02 | 0.50 | 21.04 | 24.07 | 1.41 |
> | PIGDreamer(Ours) | 18.45 | 0.92 | **14.18** | 20.86 | 2.28 | **15.76** | 18.12 | **10.56** | 2.09 |
>
> **Q3. The assumptions of the ACPOMDP framework.**
>
> Thank you for your question. In the ACPOMDPs framework, we assume that the privileged information consists of the underlying states of the environment; however, this assumption is impractical in real-world scenarios, as defining the underlying states and determining how to obtain them is challenging. Consequently, we relax this assumption to incorporate certain sensors that are available and precise, albeit costly, in real-world scenarios.
>
> For example, in the Safety-Gymnasium benchmark, we define the privileged information as including the proprioceptive state, historical actions, and the relative positions of hazards and the goal. In the Guard benchmark, the privileged information consists of the proprioceptive state and LiDAR perceptions of hazards and the goal. All the aforementioned sensors are accessible in real-world applications, making our method a viable solution for practical implementation.
>
> **Q4. Detailed Statistical Analysis.**
>
> Thank you for your suggestion. We apologize for our inability to complete all the experiments within the limited time available during the rebuttal period. Consequently, we intend to conduct a more comprehensive statistical analysis in the upcoming round of rebuttals.

---

### Official Review · Reviewer_ou54 · 2025-03-13

**Overall Recommendation:** 3

**Summary:**

The work focuses on the development of a novel safe model-based reinforcement learning method. The researchers utilize the concept of privileged information and introduce the so-called asymmetric constrained partially observable Markov decision process (ACPOMDP) task, which requires access to the actual state in addition to observations for training the critic. They utilize the basic architecture of Dreamer and extend it to function simultaneously with both observations and actual states from the environment. Two models of the world are created: a naive model and a privileged model. During the deployment phase, only the naive model with access to observations is utilized. During the learning process, a multi-component loss is employed, incorporating privileged information. Safety is ensured through standard regularization using a Lagrangian approach. Experiments are conducted using Safety Gymnasium, and the proposed method demonstrates the same performance compared to SafeDreamer, Safe-SLAC, and other privileged approaches.

## update after rebuttal
The authors have conducted additional experiments in a new Guard setting, enhancing their work's experimental part. These experiments show that PIGdreamer has low quality at a cost. These and previous results do not conclusively prove a significant advantage over other baselines for me. In my opinion, safe reinforcement learning methods should first demonstrate excellent results in terms of cost and only secondarily show good performance and high rewards. Considering my other comments, I am inclined to leave my assessment unchanged for now, but I will not object to the acceptance of the article.

**Claims And Evidence:**

The statements proposed by the authors in the article on the effectiveness of using privileged information are generally confirmed by the experiments conducted.

**Essential References Not Discussed:**

Most of the necessary papers are correctly cited.

**Experimental Designs Or Analyses:**

The experiments were conducted according to the necessary requirements, but it is difficult to say that they were convincing. PIGDreamer does not differ significantly from the baseline with privileged information in terms of reward plots and cost value. However, the disadvantage of SafeDreamer in terms of not accessing true states is obvious. Figure 4 shows that all results are within the range of variations.

**Methods And Evaluation Criteria:**

The authors use the standard Safety Gymnasium benchmark, considering many tasks there, which is the standard in this field. In general, other benchmarks could be considered, such as SafeAntMaze based on MuJoCo.

**Other Comments Or Suggestions:**

The authors were careless about the links to pictures and pictures. The order is clearly messed up. The link to page 8 doesn't lead anywhere at all. Figure 4 shows strange x-axis signatures, especially on the cost graph.

**Other Strengths And Weaknesses:**

The authors do not clearly explain how the naive and privileged world models  are separated by parameters. Based on the notation, it seems that they are trained with the same set of parameters, which is not well described. I have no other significant comments at this time.

**Questions For Authors:**

Do naive and privileged world models have the same set of parameters? How is the privileged part discarded during the deployment phase?

**Relation To Broader Scientific Literature:**

In general, the proposed idea combines well-known methods: the SafeDreamer approach with a Lagrangian, and the idea of duplicating observations and states when training a world model. The positive aspect is that the authors have successfully formalized and trained all these elements into a single model. However, the experimental results do not seem very convincing.

**Theoretical Claims:**

The paper makes a fairly obvious claim that, in the presence of complete state information, a critic will form more accurate value functions than if it has incomplete observations. However, this claim does not add any significant insight to the approach. Instead, the proof in the appendix follows the logic presented in Pineau et al. (2006).

---

> ### Author Rebuttal · Authors · 2025-04-01
>
> We sincerely appreciate the valuable suggestions provided by Reviewer OU54. In response, we have conducted additional experiments and clarifications, with the hope that these efforts will address your concerns.
>
> **Q1. Clarification Regarding Model Parameters and Deployment.**
>
> The naive and privileged world models are trained jointly using the same set of parameters. During the deployment phase, the parameters for both models are loaded; however, only the naive world model and the actor are employed to generate actions.
>
> **Q2. Relation Between Our Theoretical Claim and Proposed Method.**
>
> We appreciate your question. Our claim that a critic can develop more accurate value functions through direct access to the underlying states motivates our methodology in two key aspects:
>
> 1. **Enhanced Value Estimation.** We utilize an asymmetric architecture to facilitate more precise value estimations, thereby improving the policy.
> 2. **Mitigation of Risk Underestimation.** The analysis presented in Theorem 3.3 illustrates that reliance on partial observations results in the critic underestimating associated risks. This finding compels us to integrate privileged information into Safe Reinforcement Learning (SafeRL).
>
> **Q3. Guard Benchmark.**
>
> Thank you for your suggestion. In response, we conducted additional experiments within the Guard benchmark under a limited timeframe. The Guard benchmark is a Safe Reinforcement Learning (SafeRL) framework that includes more complex agents, such as Ant and Humanoid, as well as more challenging tasks.
>
> In our experiments, we utilized 64 x 64 pixel images as observations, along with low-dimensional sensors as privileged information. Our empirical results demonstrate that our method significantly outperforms other baseline approaches. Furthermore, we found that, in comparison to the Safety-Gymnasium benchmark, our method achieves highly competitive results within the Guard benchmark. We attribute this to the relative ease of the Safety-Gymnasium benchmark, which may obscure the advantages of our method.
>
> | Model |  | Goal_Ant_8Hazards |  |  | Goal_Ant_8Ghosts |  |  | Goal_Humanoid_8Hazards |  |
> | --- | --- | --- | --- | --- | --- | --- | --- | --- | --- |
> |  | Training time | Cost | Reward | Training time | Cost | Reward | Training time | Cost | Reward |
> | SafeDreamer | 14.27 | 1.00 | 2.54 | 16.7 | 0.93 | 10.67 | 13.94 | 10.8 | 0.34 |
> | Informed-Dreamer(Lag) | 17.03 | **0.03** | 1.32 | 18.28 | 2.54 | 10.09 | 14.03 | 10.72 | -1.60 |
> | Scaffolder(Lag) | 24.82 | 0.04 | 1.86 | 27.14 | **0.83** | 6.15 | 23.8 | 13.14 | **2.13** |
> | Distill | 20.12 | 10.52 | 1.32 | 23.21 | 1.02 | 0.50 | 21.04 | 24.07 | 1.41 |
> | PIGDreamer(Ours) | 18.45 | 0.92 | **14.18** | 20.86 | 2.28 | **15.76** | 18.12 | **10.56** | 2.09 |
>
> **Q4. The performance improvement of our method in the Safety-Gymnasium benchmark.**
>
> Thank you for your question. We acknowledge that our performance improvement in the Safety-Gymnasium benchmark may appear modest; however, there are several contributions of our work that we would like to highlight:
>
> 1. **Privileged Learning for Safety.** We investigated the application of privileged MBRL in safety-critical tasks, a topic that is rarely addressed within the SafeRL community. In this regard, we reimplemented the most advanced privileged MBRL methods, Scaffolder and Informed-Dreamer, to incorporate safety constraints. The benefits of integrating privileged information are convincingly evidenced by our experimental results.
> 2. **Efficiency.** As noted by other reviewers, our method achieves superior performance at the expense of a modest increase in training time, rendering it a practical solution for real-world applications.
> 3. **More competitive results.** In **Q3**, we discuss the empirical results from the Guard benchmark, which we hope will address your concerns regarding the performance of our method.
>
> **Q5. Other Comments.**
>
> We will update our manuscript based on your valuable suggestions. Additionally, we are in the process of developing a website for this paper, where we will include the code and videos.

---

> > ### Comment · Reviewer_ou54 · 2025-04-03
> >
> > I would like to thank the authors for providing answers to my questions and comments. I also want to mention that the authors have conducted additional experiments in a new Guard setting, which enhances the experimental part of their work. These experiments show that PIGdreamer has low quality at a cost. These and previous results do not conclusively prove a significant advantage over other baselines for me, so I will leave the score unchanged for now.

---

> > > ### Author Response · Authors · 2025-04-08
> > >
> > > We thank Reviewer ou54 for responding to our rebuttal. But we think there is a misunderstanding concerning our experiments on the Guard benchmark. To allay your concerns, we have released an anonymous repository with code and videos. https://anonymous.4open.science/r/PIGDreamer-270B.
> > >
> > > ---
> > >
> > > **Guard Benchmark.**
> > > > These experiments show that PIGDreamer has low quality at a cost.
> > > >
> > >
> > > In our experiments on the Guard benchmark, PIGDreamer incurs relatively higher costs in the Goal_Ant_8Hazards and Goal_Ant_8Ghosts tasks. However, in these tasks, PIGDreamer significantly outperforms the baselines in terms of rewards. In the Goal_Ant_8Hazards task, Informed-Dreamer (Lag) and Scaffolder (Lag) achieve the lowest costs of 0.03 and 0.04, respectively, but at the expense of extremely low rewards. In fact, the agents trained with Informed-Dreamer (Lag) and Scaffolder (Lag) develop ineffective policies, as they consistently remain stationary and fail to exhibit any movement, as evidenced by the videos attached to the anonymous repository. A similar phenomenon is observed in the Goal_Ant_8Ghosts task.
> > >
> > > Conversely, PIGDreamer learns an effective policy while maintaining costs below the threshold of 3, thereby demonstrating its significant advantages.
> > >
> > > **Reproducibility.**
> > >
> > > To address the concern regarding reproducibility, we have attached our code and videos at the following link: https://anonymous.4open.science/r/PIGDreamer-270B.
> > >
> > > **The advantages of our method.**
> > >
> > > We wish to emphasize that our method demonstrates the advantages of efficiency and high performance:
> > >
> > > - **Efficiency.**  Our method achieves optimal safety and task performance with a 23.14% increase in training time, whereas the alternative method, Scaffolder (Lag), attains the second-best performance with a 76.94% increase in training time.
> > >
> > > | Model | SafetyPointGoal2 | SafetyCarGoal1 | SafetyRacecarGoal1 | SafetyPointPush1 | SafetyPointButton1 | Avg |
> > > | --- | --- | --- | --- | --- | --- | --- |
> > > | Informed-Dreamer(Lag) | 6.42% | 1.02% | 5.98% | 9.40% | 8.01% | 6.16% |
> > > | Scaffolder(Lag) | 71.56% | 77.62% | 68.34% | 93.28% | 73.90% | 76.94% |
> > > | PIGDreamer(Ours) | 16.58% | 41.20% | 14.95% | 25.13% | 17.85% | 23.14% |
> > > - **Performance.**  PIGDreamer significantly surpasses the baselines on the Guard benchmark, effectively addressing tasks that alternative methods fail to conquer. Furthermore, in contrast to other model-based Safe Reinforcement Learning methods (SafeDreamer, LAMBDA, SAFE-SLAC), which exclusively experiment within the Safety-Gymnasium, we expand our experiments to encompass more complex agents, such as the ant and humanoid, demonstrating remarkable performance.
> > > ---
> > > We hope our response addresses your concerns and encourages you to reconsider our score.

---

### Official Review · Reviewer_byT1 · 2025-03-24

**Overall Recommendation:** 3

**Summary:**

Disclosure: I am an emergency reviewer, and also quite familiar with the paradigm of RL with privileged information.

The authors propose to exploit privileged information for policy learning in the context of safe RL. They use a world model approach like Dreamer, where an actor-critic is trained solely in a learned MDP (the world model). They use privileged information to train a privileged world model. Then, they exploit the privileged information in a variety of ways, such as aligning the privileged WM and unprivileged WM representations, using asymmetric critics and reward estimators, and generating imagined data with the privileged WM. They evaluate their method on the Safety Gymnasium benchmark, which is state-based goal-reaching tasks, and the agent must avoid hazards.

**Claims And Evidence:**

The overall system is verified by showing improvements over baselines. The improvement in returns and safety violation, compared to baselines, is marginal to moderate in my opinion, especially compared to the most competitive baseline Scaffolder. However, this method does seem more simple and faster to run which is good.

The design choice of certain components in the method are somewhat questionable, see below.

The method uses an augmented lagrangian objective to balance between reward and safety cost, but these methods are known for instability. Can the authors write more about how stable this process is,  and what they need to do to make it stable, etc.?

**Essential References Not Discussed:**

The authors do a fair job of surveying privileged reinforcement learning with world models.

The algorithm seems quite similar to Scaffolder (ICLR24), which is mentioned and compared against. But could the authors give a more detailed comparison on the similarities and differences? It seems like PIGDreamer removes some parts and replaces some objectives.

Another privileged MBRL method not mentioned is Wasserstein Believer (ICLR24).

**Experimental Designs Or Analyses:**

I am not an expert in Safe RL, but there might be some essential Safe RL baselines that are missing, like Constrained Policy Optimization, TRPO-Lagrangian, etc.? In the main paper. Luckily, I skimmed the appendix and there seems to be comparison in section D.3. I think this should be moved to the main results.

Next, for privileged reinforcement learning, a simple baseline is to train a privileged teacher with RL, and then distill it into an unprivileged student. I think this simple privileged baseline is important to show, to justify the additional complexities of using a model-based privileged RL approach.

All the environments seem to be done in low-dimensional state based environments. It is worth mentioning as a limitation, and image-based environments would be good future work.

**Methods And Evaluation Criteria:**

Twisted imagination: This procedure intends to generate trajectories with privileged information, by unrolling both privileged and unprivileged world models from the same starting state, using the same policy, and recording the pairs of state $(s^+_t, s^-_t)$. However, because each world model is stochastic, it seems unlikely that the trajectories are actually paired. Imagine an environment with a fork at the start state, and path 1 gives 100 reward while path 2 gives -100. The first WM predicts path 1, and the second WM predicts path 2.  So now your $(s^+_t, s^-_t)$ pairs should not really be paired together, since even though they are at the same timestep, they correspond to different locations in the environment.

Representation alignment: Are you pulling  both state encoders towards each other? Or is it just one directional?

Some additional baselines could be used, see below.

**Other Comments Or Suggestions:**

N/A.

**Other Strengths And Weaknesses:**

I think the application of privileged information towards solving Safe RL tasks is nicely motivated.

**Questions For Authors:**

Could you address my questions about the design choices?

Ablation experiments on the twisted imagination could be interesting, you could replace twisted imagination with the way Scaffolder does imagination?

Running a simple privileged RL baseline like distillation would be helpful for convincing readers to use your method over something simpler.

> Compared to previous works (Lambrechts et al., 2024; Hu et al., 2024) that directly reconstruct
privileged information ...this method is significantly more robust when the privileged information is excessively informative for reconstruction

I don't get what the authors are saying here, could you clarify?

There are no qualitative descriptions or videos of the tasks I can see. It would be nice to highlight strengths and weaknesses of each method's policies, by showing their rollouts and comparing them.

---
Overall: For now, I will lean on the positive side, assuming the authors take my feedback and address it.

**Relation To Broader Scientific Literature:**

It can be seen as a model-based approach to leveraging privileged information in safe RL.
Previous work in privileged RL, to my knowledge, did not focus on safe RL. Although I would be sure that people have tried exploiting privileged information for safe RL, especially in robotics.

**Theoretical Claims:**

I read the theoretical section but did not check it carefully.

---

> ### Author Rebuttal · Authors · 2025-03-31
>
> We would like to express our gratitude to Reviewer byT1 for your insightful comments.
>
> **Q1. The mismatch arises from the Twisted Imagination (TI) and the ablation of TI with the Nested Latent Imagination (NLI).**
>
> Thank you for your question. Indeed, the trajectories generated from TI exhibit variability due to their stochastic nature. To address this problem, NLI proposed in Scaffolder forces the matching between two trajectories by encoding the embedding from  $Path_2$ to $Path_1$. However, our ablation study about the NLI and TI, demonstrate that this method yields no performance improvement. We attribute this phenomenon to the robustness of the TI predictors. Although the representations of the privileged world model deteriorate during the imagination process, the predictors continue to make accurate predictions from the states $s^{*}$  and $s^{+}$ .
>
> | Task | Method | Reward | Cost |
> | --- | --- | --- | --- |
> | SafetyPointGoal2 | NLI | 10.79 | **0.41** |
> |  | **TI (Ours)** | **13.59** | 0.73 |
> | SafetyCarGoal1 | NLI | 14.79 | 0.64 |
> |  | **TI (Ours)** | **17.32** | **0.43** |
> | SafetyRacecarGoal1 | NLI | **13.99** | 1.57 |
> |  | **TI (Ours)** | 11.38 | **0.83** |
>
> **Q2. The implementation of Representation Alignment.**
>
> In our work, we pull both state encoders towards each other  using the method proposed in Dreamerv3.
>
> **Q3. Clarification regarding the privileged representation.**
>
> In our experiments, Informed-Dreamer exhibits instability in reward when the privileged information is too informative to be reconstructed from the history of partial observations. To address this issue, we enhance the state representation through **privileged representation alignment**. This approach enables us to establish correspondence between modalities (images and states), while ensuring that the training objective remains agnostic to the dimensionality of the privileged information.
>
> **Q4. Comparison with Scaffolder.**
>
> Thank you for your inquiry. We are pleased to compare our algorithm with Scaffolder (ICLR24). We will evaluate our algorithm against Scaffolder (ICLR24) in the following aspects:
>
> 1. **Theoretical Foundation:** We propose our method within the ACPOMDP framework while Scaffolder has made limited advancements in this area.
> 2. **Privileged Representations:** see **Q3**.
> 3. **Privileged Predictors:** The scaffolder comprises two groups of predictors: one group relies exclusively on the state of the naive world model, while the other depends on the state of the privileged world model. In our approach, we simplify these components by enabling the predictors to operate based on the states of all world models.
> 4. **Privileged Imagination:** see **Q1**.
> 5. **Privileged Exploration:** We eliminate this component for the following reasons:
>     1. It prolongs the training time.
>     2. The privileged actor operates using information that is inaccessible to the naive actor, potentially creating a significant disparity between their behaviors. As a result, the trajectories collected by the privileged actor become difficult for the naive actor to learn from.
> 6. **Privileged Critics:** same with Scaffolder.
> 7. **Efficiency:** Our method demonstrates enhanced efficiency in utilizing privileged information.
>
> **Q5. The teacher-student distillation baseline.**
>
> Thank you for your valuable suggestion. We have added the teacher-student distillation baseline within the limited timeframe. Our experimental results demonstrate that our method consistently outperforms the distillation baseline by a significant margin. Furthermore, we observe that, in the absence of explicit safety constraint objectives, the student fails to meet safety constraints.
>
> | Task | Model | Training time | Cost | Reward |
> | --- | --- | --- | --- | --- |
> | SafetyPointGoal2 | Distill | 30.02 | 4.54 | 6.59 |
> |  | **Ours** | **29.71** | **1.31** | **11.61** |
> | SafetyCarGoal1 | Distill | 29.21 | 3.08 | 11.74 |
> |  | **Ours** | **28.87** | **0.93** | **17.37** |
> | SafetyRacecarGoal1 | Distill | **30.19** | 3.47 | 10.21 |
> |  | **Ours** | 31.03 | **1.17** | **10.99** |
> | SafetyPointPush1 | Distill | **31.03** | 12.37 | 12.26 |
> |  | **Ours** | 34.87 | **1.15** | **17.10** |
> | SafetyPointButton1 | Distill | 30.02 | 11.12 | **6.77** |
> |  | **Ours** | **29.54** | **2.01** | 5.97 |
>
> **Q6. Experimental setup.**
>
> Thank you for your question. Our experiments actually use 64 × 64 pixel images for agent observations, while using low-dimensional state as privileged information. We will clarify this experimental setup in the final version of our manuscript.
>
> **Q7. Wasserstein Believer.**
>
> We plan to incorporate a discussion of its relevance in the next version of our paper.
>
> **Q8. The stability of the augmented lagrangian method.**
>
> We regret our inability to engage in further discussion on this matter due to space limitations. Given that the augmented Lagrangian method is a widely accepted practice within the Safe RL community, we have not investigated its stability.

---

### Decision · Program_Chairs · 2025-05-01

**Decision:**

Accept (poster)

**Comment:**

The paper presents PIGDreamer, a model-based reinforcement learning approach that leverages privileged information during training to enhance safety and performance in partially observable environments. The authors introduce the Asymmetric Constrained Partially Observable Markov Decision Process (ACPOMDP) framework, which allows the value function to access underlying states, improving policy learning. PIGDreamer incorporates privileged information through privileged representation alignment, asymmetric actor-critic structures, and enhanced reward prediction mechanisms. The empirical results, tested on the Safety-Gymnasium benchmark, indicate that the approach achieves superior safety and task-centric performance compared to existing methods.

The paper proposes a novel approach to incorporate privileged information for safe RL, provides a theoretical foundation for privileged RL, and has promising experimental results in terms of safety, task performance, and efficiency compared to baselines.

The gains are small, the experimental benchmark tasks are somewhat non-standard, and the qualitative presentation of results is poor. Some of these were addressed in the rebuttals, leading to overall positive scores from each reviewer. The authors are encouraged to incorporate these changes and new results into future versions.